# Antimicrobial Activity and Toxicity of Newly Synthesized 4-[4-(benzylamino)butoxy]-9*H*-carbazole Derivatives

**DOI:** 10.3390/ijms241813722

**Published:** 2023-09-06

**Authors:** Katarzyna Niedziałkowska, Aleksandra Felczak, Iwona E. Głowacka, Dorota G. Piotrowska, Katarzyna Lisowska

**Affiliations:** 1Department of Industrial Microbiology and Biotechnology, Faculty of Biology and Environmental Protection, University of Lodz, 90-237 Lodz, Poland; aleksandra.felczak@biol.uni.lodz.pl (A.F.); katarzyna.lisowska@biol.uni.lodz.pl (K.L.); 2Bioorganic Chemistry Laboratory, Faculty of Pharmacy, Medical University of Lodz, 90-151 Lodz, Poland; iwona.glowacka@umed.lodz.pl (I.E.G.); dorota.piotrowska@umed.lodz.pl (D.G.P.)

**Keywords:** carbazole derivatives, antibacterial agents, antifungal potential

## Abstract

One of the main challenges of medicinal chemistry is the search for new substances with antimicrobial potential that could be used in the fight against pathogenic microorganisms. Therefore, the antimicrobial activity of newly synthesized compounds is still being investigated. Carbazole-containing compounds appear to be promising antibacterial, antifungal, and antiviral agents. The aim of this study was to examine the antimicrobial potential and toxicity of newly synthesized isomeric fluorinated 4-[4-(benzylamino)butoxy]-9*H*-carbazole derivatives. Their antimicrobial activity against bacteria and fungi was tested according to CLSI guidelines. Similarly to previously studied carbazole-containing compounds, the tested derivatives showed the ability to effectively inhibit the growth of Gram-positive bacteria. The addition of carbazole derivatives **2**, **4**, and **8** at the concentration of 16 µg/mL caused the inhibition of *S. aureus* growth by over 60%. The MIC value of compounds **2**–**5** and **7**–**10** was 32 µg/mL for *Staphylococcus* strains. Gram-negative strains of *E. coli* and *P. aeruginosa* were found to be more resistant to the tested carbazole derivatives. *E. coli* cells treated with compounds **3** and **8** at a concentration of 64 µg/mL resulted in a greater-than-40% reduction in bacterial growth. In the case of the *P. aeruginosa* strain, all compounds in the highest concentration that we tested limited growth by 35–42%. Moreover, an over-60% inhibition of fungal growth was observed in the cultures of *C. albicans* and *A. flavus* incubated with 64 µg/mL of compounds **2** or **7** and **1** or **4**, respectively. The hemolysis of red blood cells after their incubation with the tested carbazole derivatives was in the range of 2–13%. In the case of human fibroblast cells, the toxicity of the tested compounds was higher. Derivative **1**, functionalized with fluorine in position 2 and its hydrobromide, was the least toxic. The obtained results indicated the antimicrobial potential of the tested 4-[4-(benzylamino)butoxy]-9*H*-carbazole derivatives, especially against *S. aureus* strains; therefore, it is worth further modifying these structures, in order to enhance their activity against pathogenic microorganisms.

## 1. Introduction

Despite the availability of a wide range of antibiotics, infectious diseases remain a challenge for clinical medicine, and their effective treatment is a public health priority. It is estimated that, in 2017, there were over 10 million deaths due to drug-resistant pathogens and sepsis, meaning that microorganisms accounted for more than 20% of deaths worldwide. In low- and middle-income countries, where mortality from communicable diseases is particularly high, efforts should be made to prevent and treat them more effectively. That is also important from the perspective of limiting the spread of drug-resistant microorganisms. Among the top five most dangerous bacteria associated with human deaths, *Staphylococcus aureus*, *Escherichia coli*, *Streptococcus pneumoniae*, *Klebsiella pneumoniae*, and *Pseudomonas aeruginosa* are most often mentioned. These pathogens were linked to over 30% of deaths from infection-related diseases in 2019 [1].

Carbazole is a tricyclic nitrogen-containing heteroaromatic compound, recognized as one of the most promising pharmacophores in medicinal chemistry, due to the various biological properties of carbazole-containing compounds. Carbazole derivatives have been found among natural products, such as plant alkaloids (murrayanine), as well as synthetic products, including carvedilol **A** (Figure 1), which is an adrenoceptor antagonist. The antimicrobial potency of carbazole-containing compounds has been repeatedly emphasized in the literature [2].

In our previous study, the antibacterial activity of carvedilol **A** was examined against Gram-positive and Gram-negative bacteria. The tested compound **A** showed the strongest properties against Gram-positive strains [3]. These results prompted us to search for further new carbazole analogues with antibacterial and antifungal potential. The synthesis of 4-[4-(benzylamino)butoxy]-9*H*-carbazole **B** (Figure 1) was carried out, and its antimicrobial activity and toxicity were assessed. Similarly to carvedilol **A**, the tested derivative **B** showed promising activity against Gram-positive bacteria [4].

In the continuation of our studies on carbazole-containing compounds, a new series of derivatives (Figure 2) was designed, via the modification of the benzylamine substituent in the previously described compound **B**, so that we could test their antimicrobial properties. As the solubility of carbazole derivatives is an important issue, in addition to carbazole derivatives containing functionalized benzylamine moieties (compounds **1**–**5**), their respective hydrobromides (compounds **6**–**10**) were prepared and studied. Their action against bacterial cells was examined, as well as their action against pathogenic fungal strains. Toxicity studies into the obtained derivatives were carried out, using human fibroblasts and red blood cells.

## 2. Results

To obtain the designed compounds **1**–**10**, 4-(4-bromobutoxy)-9*H*-carbazole **11** was synthesized from commercially available 4-hydroxycarbazole via the application of the procedure previously described in the literature [5], and was subsequently transformed into carbazole derivatives **1**–**5**, with functionalized benzylamine moieties or their respective hydrobromides **6**–**10** (Figure 1). The structures and purity of all the synthesized compounds were determined via ^1^H and ^13^C NMR spectra, as well as via elemental analyses.

The antimicrobial properties of 4-[4-(benzylamino)butoxy]-9*H*-carbazole derivatives **1**–**10** were examined using Gram-positive *S. aureus* and *S. epidermidis* strains, as well as Gram-negative *E. coli* and *P. aeruginosa* bacteria. Their antifungal activity was also evaluated against *C. albicans* yeast and *A. flavus* filamentous fungi. Among the tested microorganisms, the strongest antimicrobial action of the carbazole derivatives **1**–**10** was found against Gram-positive bacteria, and *S. aureus* appeared to be more sensitive than *S. epidermidis* (Figure 3A,B). The addition of compounds **2**, **3**, **4**, **8**, or **9** at a concentration of 16 µg/mL caused an over-50% growth reduction in of *S. aureus*. In the *S. aureus* and *S. epidermidis* cultures, the values of the minimum inhibitory concentrations (MICs) reached 32 µg/mL for compounds **2**, **3**, **4**, **5**, **7**, **8**, **9**, and **10** (Table 1). Compounds **1** and **6** were less active against Gram-positive strains, and their MIC value was 64 µg/mL. The minimum bactericidal concentrations (MBCs) of compounds **2** and **4** in *S. aureus* and *S. epidermidis* were 32 µg/mL. The MBCs for compounds **5**, **7**, **8**, **9**, and **10** reached the value of 64 µg/mL, and for compounds **1**, **3**, and **6**, they were above the highest tested concentrations. Moreover, ciprofloxacin and amphotericin B were tested as the positive control compounds in an antimicrobial assay. The MICs of ciprofloxacin for *S. aureus*, *S. epidermidis*, *E. coli*, and *P. aeruginosa* reached the values of 0.5, 0.25, 0.125, and 0.5 µg/mL, respectively. In the case of *A. flavus* and *C. albicans*, the MICs of amphotericin B were 4 and 1 µg/mL, respectively.

In the case of the *E. coli* and *P. aeruginosa* strains, a growth inhibition of approximately 40% was observed in cultures supplemented with the tested compounds at the concentration of 64 µg/mL (Figure 4A,B). The strongest antimicrobial properties against *E. coli* were exhibited by compounds **3** and **8**, while the weakest antimicrobial activity was shown by compound **6**. In the case of *P. aeruginosa*, the growth inhibition caused by all the tested compounds added to the bacterial cultures at the concentrations of 32 and 64 µg/mL occurred at a similar level. The MBC values for Gram-negative strains were above the highest tested concentration.

The addition of 4-[4-(benzylamino)butoxy]-9*H*-carbazole derivatives at the concentration of 64 µg/mL to the cultures of yeast and filamentous fungi resulted in a significant growth inhibition in the microorganisms (Figure 5A,B). The strongest antifungal activity against *A. flavus* was noted in samples supplemented with compounds **1** and **4**, where a growth reduction of over 65% was observed. *C. albicans* was the most sensitive to the action of compounds **2** and **7**. Their addition to yeast cultures resulted in growth inhibition at the level of 60%. The weakest antifungal properties against yeast were found for compound **6**, which caused only a 15% growth limitation at the highest tested concentration.

The hemolytic potential of the tested 4-[4-(benzylamino)butoxy]-9*H*-carbazole derivatives was determined at the concentration range of 1–64 µg/mL (Figure 6). The hemolysis was calculated after 24 h of treatment of red blood cells with the tested compounds. The hemolytic activity in the samples supplemented with the tested 4-[4-(benzylamino)butoxy]-9*H*-carbazole derivatives did not reach values greater than 13%, so they did not have significant effects on the red blood cells.

The cytotoxic activity of ten 4-[4-(benzylamino)butoxy]-9*H*-carbazole derivatives was studied in the same concentration range as hemolysis (Figure 7). The tested compounds were more toxic to human fibroblasts than to erythrocytes. The viability of fibroblasts treated with all compounds at the highest tested concentration was reduced by over 90%. The highest toxicity was detected in samples supplemented with compound **9**. Its addition at the concentration of 4 µg/mL caused a decrease in cell viability of 40%. The least toxic agent was compound **1**, the addition of which, at a concentration of 32 µg/mL, reduced the cell viability by about 60%.

## 3. Discussion

The increasing mortality due to infectious diseases caused by drug-resistant microorganisms has led the design of new molecules with antimicrobial properties to be an important branch of medicinal chemistry. Carbazole-containing compounds, such as natural carbazole alkaloids obtained from the leaves of *Murraya koenigii*, have demonstrated notable antibacterial and antifungal activity. The tested alkaloids have exhibited the ability to inhibit growth in the *Escherichia*, *Staphylococcus*, *Streptococcus,* and *Candida* strains. The MICs of these substances against *S. aureus* and *S. pyogenes* reached 25 µg/mL [2]. Methods for the synthesis of various substances containing a carbazole ring with antimicrobial properties are regularly described in the literature [6,7,8]. There is also evidence that commonly available drugs containing carbazole in the structure may have antibacterial activity. The popular adrenoceptor antagonist carvedilol, which has carbazole in the structure, has shown antibacterial potential against Gram-positive strains, whereas its action toward Gram-negative bacteria was weak [3]. Additionally, an enhancement in ciprofloxacin activity toward *S. aureus* was noted in cultures incubated simultaneously with an antibiotic and carvedilol [9]. Several papers have indicated the high antimicrobial potential of carbazole-containing compounds [6,7,8]. Xie et al. [8] synthesized different carbazole–oxadiazole and tetrazole analogues with great antibacterial potential, especially against methicillin-resistant *S. aureus* (MRSA) and *P. aeruginosa* strains. The authors indicated that the enhancement in antimicrobial activity was achieved as a result of the introduction of a cycloalkyl moiety to the tested molecules. It has also been shown that the introduction of a fluorophenyl group onto position 4 of carbazole can increase its antibacterial potential, especially against the *P. aeruginosa* strain. Due to the activity of the efflux pump relative to small molecules, docking simulations have considered the interactions of carbazole–oxadiazoles and carbazole–tetrazoles with the outer-membrane porins of *P. aeruginosa* cells. Although the tested compounds may be effective agents against *P. aeruginosa*, the total score showed no variation or regularity. Therefore, other factors should be considered, and this issue still requires further research [8]. Other studies have also confirmed the high antimicrobial potential of carbazole–oxadiazole derivatives against *S. aureus*. Docking simulations indicated the interaction of the carbazole–oxadiazole derivative with gyrase in the *S. aureus* strain. The nitrogen atom and sulfhydryl residues of the tested compound created hydrogen bonds with amino acid groups in the gyrase pocket. Moreover, carbazole and oxadiazole rings may interact hydrophobically or via Van der Waals forces with residues located on the outside of the binding pocket [7]. On the other hand, ursolic acid derivatives containing a carbazole moiety have been found to exhibit both a high antimicrobial and a high antitumor potential. Among the tested compounds, the most active were those substituted with 5-fluoroindole and N-(dimethylamino)propyl amide chains. The MICs of these derivatives reached values from 3.9 to 31.2 µg/mL [6]. The introduction of a phenyl or thiazole ring to the carbazole structure can improve the antimicrobial activity [10]. Moreover, a significant antimicrobial activity of N-substituted 1H-dibenzo[a,c]carbazole has been observed, compared with the action of some antibiotics, including ketoconazole and amikacin [11]. Salih et al. [12] synthesized 9*H*-carbazole-based azole derivatives, and demonstrated their diverse antimicrobial potential. Among the tested compounds, derivative 8 ((4-[3-(9*H*-Carbazol-9-yl-acetyl)triazanylidene]-5-methyl-2-phenyl-2,4-dihydro-3H-pyrazol-3-one)) showed the highest activity against *S. aureus* and *E. coli* strains, and the MIC values reached 1.1 and 6.4 µg/ml, respectively. This compound also exhibited moderate potential against fungal cells. The MICs for *C. albicans* and *A. fumigatus* were 9.6 and 10.3, respectively [13]. High antimicrobial potential was also shown in the case of newly synthesized carbazole N-phenylacetamide hybrids. Methyl2-(6-chloro-9-(2-oxo-2-(phenylamino)ethyl)-9Hcarbazol-2-yl) propanoate derivatives exhibited antibacterial activity against Gram-positive *S. aureus* and *Bacillus subtilis* strains, as well as toward Gram-negative bacteria E. coli and P. aeruginosa, with MICs ranging from 0.25 to 8 µg/ml. Moreover, the compounds showed anti-inflammatory and antioxidant activities. A molecular docking study revealed that they were able to bind to receptor 2MBR. Compounds 3g, 3i, and 3h interacted with the active site of the MurB enzyme receptor in *E. coli* cells, and demonstrated a stronger capability to inhibit this receptor, compared to ciprofloxacin [14]. Merzouki at al. [15] synthesized three new carbazole derivatives (DC1–DC3) in an eco-friendly manner, and showed their antibacterial and antifungal activity. The introduction of a triazole moiety to DC3 structures increased its pharmacokinetic and antifungal activity compared to DC1 (dimethyl pyrazole) and DC2 (pyrazole) derivatives. The MICs of compounds DC1, DC2, and DC3 against *C. albicans* were 0.031, 0.031, and 0.016 mg/ml, respectively. Molecular docking studies revealed the interaction between the obtained derivatives and the active site of the structure of topoisomerase IV (3FV5) in *E. coli* cells. The binding energy values of DC1, DC2, DC3, and streptomycin were 6.5, 6.2, 6.1, and 5.3 kcal/mol, respectively [15]. Ding et al. [16] received novel carbazole derivatives containing a 3-(piperazin-1-yl) propan-2-ol moiety, and evaluated their potential against plant bacterial pathogens. The EC50 of compound A9 against *Xanthomonas oryzae* and *Xanthomonas axonopodis* demonstrated the values of 6.80 and 6.37 µg/ml, respectively. It was found that compound A9 showed an ability to damage the bacterial cell membrane, which resulted in a leak of cytoplasmic components. At the same time, this compound was characterized by a low phytotoxicity, which could enable its practical use in plant protection against bacterial pathogens [16].

The promising results of the antimicrobial activity of 4-(4-(benzylamino)butoxy)-9*H*-carbazole [3] prompted us to synthesize its isomeric fluorinated derivatives **1**–**10**. The unusual nature of fluoride affects the properties of the drug molecules into which it is introduced. It can enhance their metabolic stability or change their reactivities and physical properties [13]. The obtained results indicated that the tested fluorinated derivatives of 4-[4-(benzylamino)butoxy]-9*H*-carbazole may demonstrate better properties against *S. aureus* cells than the parent compound. Moreover, a relationship between the position of the introduced fluorine atom and the activity of the synthesized products was observed. The best antibacterial activity was exhibited by compound **8**, substituted with fluoride at position 4, the addition of which, at a concentration of 16 µg/mL, inhibited *S. aureus* growth by 65%, whereas the parent compound, at the concentration of 20 µg/mL, caused only a 22% reduction in bacterial growth. In the case of compound **6**, substituted with fluoride at position 2, a reduction in antibacterial properties was noted. The MIC values of compound **6** and the parent substance were 64 and 30 µg/mL, respectively. All tested derivatives showed inferior properties against the Gram-negative bacterium *P. aeruginosa*, while compound **8** inhibited the growth of *E. coli* by 50% at the highest tested concentration, compared to the parent compound, which inhibited growth by just 10% [4].

Various salts of active pharmaceutical ingredients (APIs) are often synthesized to achieve a better solubility, stability, or activity, as well as a lower toxicity. It is estimated that approximately 50% of approvals issued by the US FDA concern preparations containing APIs in the form of salts. Half of the top 200 prescription pharmaceuticals in the USA contain the API in the salt form. Bromides are one of the available salts [17], and were used in our work (compounds **6**–**10**). The improved solubility of carbazole derivatives prompted us to investigate the activity and toxicity of their hydrobromides. The obtained results indicated slight differences in the activity and toxicity of compounds **1–5** and their bromides, which indicated the value of searching for other salts that could have a positive effect on the physicochemical and biological properties of the tested compounds.

Evaluation of the hemolytic activity of ten 4-(4-(benzylamino)butoxy)-9*H*-carbazole derivatives did not show significant effects after the treatment of red blood cells with the tested compounds. The hemolysis level ranged from several percent to over twelve percent after 24 h of incubation of erythrocytes with the highest tested concentration of the derivatives. The obtained results corresponded with the data in the literature available for carbazole–oxadiazoles [7,8]. For example, the hemolytic rates of oxadiazolethiol–carbazole derivatives were calculated in the range of 0.5–2.5%, depending on the concentrations of the compounds. Only one derivative caused about 20% hemolysis in samples incubated with the addition of the highest tested concentration of 32 µg/mL [8]. Carbazole derivatives with antimicrobial activity showed different degrees of cytotoxicity against various cell lines. Oxadiazolethiol– and tetrazole–carbazole derivatives demonstrated a slight cytotoxic potential. Their half-maximal inhibitory concentration (IC_50_) toward Hek 293 T cells was above 64 µg/mL [8]. In our study, the IC_50_ values of ten 4-[4-(benzylamino)butoxy]-9*H*-carbazole derivatives were between 8 and 32 µg/mL for human fibroblasts. In the cases of carbazole derivatives containing aminoguanidine or triazine in their structures, the IC_50_ values for gastric cancer cells and human normal hepatic cells were within the range of 0.4–42.1 µg/mL [18]. The concentrations for carbazole-containing ursolic acid derivatives effective against hepatocarcinoma cell lines were in the range of 0.6–100 µg/mL [6].

## 4. Materials and Methods

General information: The ^1^H NMR spectra were taken in CDCl_3_ or CD_3_OD using a Bruker Avance III (600 MHz); the chemical shifts δ were given in ppm, with respect to the TMS and coupling constant *J* in Hz. The ^13^C NMR spectra were recorded in a ^1^H-decoupled mode for CDCl_3_ or CD_3_OD solutions using the Bruker Avance III (600 MHz) spectrometer, at 151 MHz. The IR spectral data were measured on a Bruker Alpha-T FT-IR spectrometer. The melting points were determined via a Boetius apparatus, and were left uncorrected. The elemental analyses were performed at the Microanalytical Laboratory of the Faculty of Pharmacy (Medical University of Lodz) on a Perkin Elmer PE 2400 CHNS analyzer, and their results were found to be in good agreement (±0.3%) with the theoretical values.

### 4.1. Synthesis of 4-[4-(benzylamino)butoxy]-9H-carbazole Derivatives 1–10

#### 4.1.1. General Procedure for the Preparation of Carbazole Derivatives 1–5

To a solution of 4-(4-bromobutoxy)-9*H*-carbazole **11** (1.00 mmol) in dry acetonitrile (15 mL), respectively, benzylamine (1.20 mmol), triethylamine (1.20 mmol), and KI (1.20 mmol) were added, and the mixture was heated under reflux for 5h. After that, the suspension was filtered off, and the residue was concentrated. The crude product was purified via chromatography using a silica gel column, with a dichloromethane–methanol mixture (100:1 to 50:1 *v*/*v*), to produce the respective carbazole derivatives **1**–**5**.

4-[4-(2-fluorophenylmethylamino)butoxy]-9*H*-carbazole **1**

From carbazole **11** (0.16 mmol), compound **1** (0.016 g, 28%) was obtained as an amorphous yellowish solid; m.p. 179–181 °C. IR (KBr, cm^−1^) ν_max_: 3424, 2925, 2854, 1626, 1607, 1587, 1491, 1453, 1440, 1261, 1230, 1095, 1036, 786, 751, 724. ^1^H NMR (600 MHz, CD_3_OD): δ = 8.25 (d, *J* = 7.8 Hz, 1H), 7.45–7.39 (m, 3H), 7.36–7.32 (m, 1H), 7.30 (t, *J* = 7.3 Hz, 1H), 7.19–7.13 (m, 3H), 7.07 (d, *J* = 8.0 Hz, 1H), 6.67 (d, *J* = 7.9 Hz, 1H), 4.28 (t, *J* = 5.5 Hz, 2H), 4.09 (s, 2H), 3.03 (t, *J* = 7.8 Hz, 2H), 2.09–1.97 (m, 4H). ^13^C NMR (151 MHz, CD_3_OD): δ = 161.25 (C_1_, *J* = 246.4 Hz), 155.18, 141.56, 139.33, 131.31 (C_3_, *J* = 3.3 Hz), 130.63, 126.05, 124.43, 124.19, 122.28, 121.47 (C_2_, *J* = 14.8 Hz), 118.33, 115.22 (C_6_, *J* = 21.2 Hz), 112.08, 109.75, 103.43, 100.00, 78.09, 66.89, 44.89 (C_-CH2-_ *J* = 3.4 Hz), 26.57, 24.35. Anal. calcd. for C_23_H_23_FN_2_O × 0.5H_2_O: C, 74.37; H, 6.24; N, 7.54. Found: C, 74.43; H, 6.16; N, 7.61.

4-[4-(3-fluorophenylmethylamino)butoxy]-9*H*-carbazole **2**

From carbazole **11** (0.32 mmol), compound **2** (0.030 g, 53%) was obtained as a yellowish oil. IR (film, cm^−1^) ν_max_: 3408, 2937, 2868, 2804, 1626, 1605, 1586, 1506, 1454, 1440, 1346, 1261, 1213, 1129, 1093, 784, 752, 719. ^1^H NMR (600 MHz, CDCl_3_): δ = 8.30 (d, *J* = 7.9 Hz, 1H), 8.07 (s, 1H), 7.42–7.36 (m, 2H), 7.32 (t, *J* = 8.0 Hz, 1H), 7.29–7.24 (m, 1H), 7.24–7.21 (m, 1H), 7.09 (d, *J* = 7.6 Hz, 1H), 7.06 (d, *J* = 2.2 Hz, *J* = 9.8 Hz, 1H), 7.03 (d, *J* = 8.1 Hz, 1H), 6.93 (d, *J* = 2.2 Hz, *J* = 7.7 Hz, 1H), 6.66 (d, *J* = 7.9 Hz, 1H), 4.25 (t, *J* = 6.3 Hz, 2H), 3.82 (s, 2H), 2.78 (t, *J* = 7.1 Hz, 2H), 2.08–2.03 (m, 2H), 1.89–1.83 (m, 2H). ^13^C NMR (151 MHz, CDCl_3_): δ = 163.02 (C_1_, *J* = 245.5 Hz), 155.61, 143.16 (C_5_, *J* = 7.3 Hz), 140.94, 138.70, 129.82 (C_3_, *J* = 7.9 Hz), 126.70, 124.91, 123.61 (C_4_, *J* = 2.2 Hz), 123.03, 122.77, 119.63, 114.88 (C_6_, *J* = 20.8 Hz), 113.78 (C_2_, *J* = 21.6 Hz), 112.70, 109.93, 103.36, 101.08, 67.73, 53.44, 49.09, 27.33, 26.88. Anal. calcd. for C_23_H_23_FN_2_O × 0.25H_2_O: C, 75.28; H, 6.46; N, 7.64. Found: C, 75.20; H, 6.45; N, 7.60.

4-[4-(4-fluorophenylmethylamino)butoxy]-9*H*-carbazole **3**

From carbazole **11** (0.32 mmol), compound **3** (0.030 g, 53%) was obtained as a yellowish oil. IR (film, cm^−1^) ν_max_: 3387, 3287, 2932, 2808, 2781, 1602, 1508, 1452, 1440, 1349, 1332, 1261, 1215, 1160, 1096, 836, 824, 786, 754, 727. ^1^H NMR (600 MHz, CDCl_3_): δ = 8.30 (d, *J* = 7.9 Hz, 1H), 8.07 (s, 1H), 7.42–7.36 (m, 2H), 7.32 (t, *J* = 7.9 Hz, 1H), 7.29–7.26 (m, 2H), 7.24–7.20 (m, 1H), 7.03 (d, *J* = 8.2 Hz, 1H), 7.01–6.96 (m, 2H), 6.65 (d, *J* = 7.9 Hz, 1H), 4.25 (t, *J* = 6.3 Hz, 2H), 3.78 (s, 2H), 2.77 (t, *J* = 7.1 Hz, 2H), 2.08–2.02 (m, 2H), 1.88–1.82 (m, 2H). ^13^C NMR (151 MHz, CDCl_3_): δ = 161.91 (C_1_, *J* = 242.2 Hz), 155.68, 140.93, 138.70, 136.15 (C_4_, *J* = 3.3 Hz), 129.66 (C_3_, *J* = 8.0 Hz), 126.70, 124.90, 123.03, 122.77, 119.61, 115.16 (C_2_, *J* = 19.4 Hz), 112.71, 109.94, 103.35, 101.07, 67.72, 53.25, 49.07, 27.35, 26.87. Anal. calcd. for C_23_H_23_FN_2_O: C, 76.22; H, 6.40; N, 7.73. Found: C, 76.30; H, 6.55; N, 7.68.

(*R*)-4-[4-(1-phenylethylamino)butoxy]-9*H*-carbazole **4**

From carbazole **11** (0.16 mmol), compound (*R*)-**4** (0.035 g, 62%) was obtained as a yellowish oil. IR (film, cm^−1^) ν_max_: 3407, 3287, 2934, 2786 1626, 1604, 1585, 1507, 1454, 1440, 1385, 1347, 1333, 1304, 1262, 1214, 1096, 786, 754, 723, 700. ^1^H NMR (600 MHz, CDCl_3_): δ = 8.21 (d, *J* = 7.7 Hz, 1H), 8.04 (s, 1H), 7.46–7.42 (m, 2H), 7.38–7.35 (m, 2H), 7.31 (ddd, *J* = 2.7 Hz, *J* = 5.5 Hz, *J* = 7.9 Hz 1H), 7.29–7.27 (m, 3H), 7.24 (t, *J* = 8.0 Hz, 1H), 6.97 (d, *J* = 8.0 Hz, 1H), 6.47 (d, *J* = 8.0 Hz, 1H), 4.20–4.14 (m, 1H), 4.01–3.91 (m, 2H), 2.82–2.70 (m, 2H), 2.24–2.09 (m, 2H), 1.89–1.74 (m, 5H). ^13^C NMR (151 MHz, CDCl_3_): δ = 155.12, 140.86, 138.64, 137.21, 129.24, 128.97, 127.72, 126.68, 124.96, 123.15, 122.54, 120.05, 112.49, 109.97, 103.61, 101.12, 66.98, 59.23, 46.06, 26.86, 23.89, 21.18. Anal. calcd. for C_24_H_26_N_2_O: C, 80.41; H, 7.31; N, 7.81. Found: C, 80.28; H, 7.26; N, 7.77.

(*S*)-4-[4-(1-phenylethylamino)butoxy]-9*H*-carbazole **5** (enantiomer of **4**)

From carbazole **11** (0.16 mmol), compound (*S*)-**5** (0.035 g, 62%) was obtained as a yellowish oil. Anal. calcd. for C_24_H_26_N_2_O×H_2_O: C, 76.56; H, 7.50; N, 7.44. Found: C, 76.48; H, 7.61; N, 7.33.

#### 4.1.2. General Procedure for the Preparation of Hydrobromides **6**–**10**

To a solution of 4-(4-bromobutoxy)-9*H*-carbazole **11** (1.00 mmol) in dry acetonitrile (15 mL), benzylamine (1.5 mmol) and KI (1.50 mmol) were added, and the mixture was heated under reflux for 5 h. After that, the suspension was filtered off, and the residue was concentrated. Diethyl ether was added, and the solid was precipitated, to produce the respective hydrobromides **6**–**10**.

The obtained compounds were analyzed using a standard analytical test for the presence of bromine and/or iodine anions: the solution of the tested compound was treated with concentrated sulfuric acid. Then, chloroform was added dropwise, followed by chlorine water. The organic layer turned brownish, proving the presence of the respective hydrobromides **6**–**10** only [19].

4-[4-(2-fluorophenylmethylamino)butoxy]-9*H*-carbazole hydrobromide **6**

From carbazole **11** (0.32 mmol), compound **6** (0.098 g, 70%) was obtained as an amorphous yellowish solid; m.p. 180–182 °C. IR (KBr, cm^−1^) ν_max_: 3396, 3301, 2932, 2816, 1617, 1603, 1586, 1452, 1440, 1349, 1284, 1214, 1097, 955, 786, 749, 726. ^1^H NMR (600 MHz, CD_3_OD): δ = 8.24 (d, *J* = 7.7 Hz, 1H), 7.52–7.48 (m, 2H), 7.44 (d, *J* = 8.1 Hz, 1H), 7.37–7.33 (m, 1H), 7.31 (t, *J* = 8.0 Hz, 1H), 7.26–7.22 (m, 2H), 7.17–7.14 (m, 1H), 7.08 (d, *J* = 8.1 Hz, 1H), 6.70 (d, *J* = 7.9 Hz, 1H), 4.36–4.32 (m, 2H), 4.31 (s, 2H), 3.30–3.25 (m, 2H), 2.12–2.04 (m, 4H). ^13^C NMR (151 MHz, CD_3_OD): δ = 161.28 (C_1_, *J* = 247.7 Hz), 155.05, 141.58, 139.35, 131.95, 131.92, 131.90, 126.07, 124.77 (C_3_, *J* = 3.3 Hz), 124.23, 122.22 (C_2_, *J* = 12.2 Hz), 118.35, 118.24, 115.54 (C_6_, *J* = 21.7 Hz), 112.04, 109.79, 103.54, 100.02, 66.66, 47.36, 44.23 (C_-CH2-_ = 4.1 Hz), 26.23, 23.23. Anal. calcd. for C_23_H_24_BrFN_2_O×H_2_O: C, 59.88; H, 5.68; N, 6.07. Found: C, 59.74; H, 5.53; N, 5.96.

4-[4-(3-fluorophenylmethylamino)butoxy]-9*H*-carbazole hydrobromide **7**

From carbazole **11** (0.32 mmol), compound **7** (0.070 g, 62%) was obtained as an amorphous yellowish solid; m.p. 181–183 °C. IR (KBr, cm^−1^) ν_max_: 3394, 3048, 2925, 2772, 2741, 2526, 1599, 1584, 1452, 1347, 1331, 1259, 1214, 1150, 1096, 938, 860, 785, 755, 725, 682. ^1^H NMR (600 MHz, CD_3_OD): δ = 8.25 (d, *J* = 7.7 Hz, 1H), 7.47–7.43 (m, 2H), 7.37–7.33 (m, 1H), 7.31 (t, *J* = 8.0 Hz, 1H), 7.28–7.24 (m, 2H), 7.22–7.18 (m, 1H), 7.17–7.14 (m, 1H), 7.08 (d, *J* = 8.2 Hz, 1H), 6.71 (d, *J* = 7.9 Hz, 1H), 4.37–4.32 (m, 2H), 4.23 (s, 2H), 3.28–3.22 (m, 2H), 2.15–2.08 (m, 4H). ^13^C NMR (151 MHz, CD_3_OD): δ = 162.91 (C_1_, *J* = 261.2 Hz), 155.11, 141.57, 139.35, 133.53 (C_5_, *J* = 7.6 Hz), 130.82 (C_3_, *J* = 8.6 Hz), 126.10, 125.54 (C_4_, *J* = 3.4 Hz), 124.26, 122.29, 122.17, 118.40, 116.37 (C_2_, *J* = 23.0 Hz), 116.12 (C_6_, *J* = 21.1 Hz), 112.04, 109.82, 103.54, 100.07, 66.66, 50.23, 47.32, 26.22, 23.28. Anal. calcd. for C_23_H_24_BrFN_2_O × H_2_O: C, 59.88; H, 5.68; N, 6.07. Found: C, 60.01; H, 5,63; N, 6.14.

4-[4-(4-fluorophenylmethylamino)butoxy]-9*H*-carbazole hydrobromide **8**

From carbazole **11** (0.16 mmol), compound **8** (0.058 g, 83%) was obtained as an amorphous yellowish solid; m.p. 203–206 °C. IR (KBr, cm^−1^) ν_max_: 3419, 3314, 2951, 2813, 1626, 1603, 1507, 1452, 1440, 1349, 1260, 1214, 1095, 952, 835, 786, 753, 726. ^1^H NMR (600 MHz, CD_3_OD): δ = 8.25 (d, *J* = 7.8 Hz, 1H), 7.46–7.43 (m, 3H), 7.35 (td, *J* = 1.1 Hz, *J* = 7.2 Hz, 1H), 7.31 (t, *J* = 8.0 Hz, 1H), 7.16 (t, *J* = 7.9 Hz, 1H), 7.13–7.10 (m, 2H), 7.08 (t, *J* = 8.1 Hz, 1H), 6.69 (d, *J* = 7.9 Hz, 1H), 4.31 (t, *J* = 5.3 Hz, 2H), 4.11 (s, 2H), 3.14 (t, *J* = 8.4 Hz, 2H), 2.12–2.04 (m, 4H). ^13^C NMR (151 MHz, CD_3_OD): δ = 163.18 (C_1_, *J* = 247.7 Hz), 155.06, 141.58, 139.35, 131.65 (C_3_, *J* = 8.7 Hz), 128.38 (C_4_, *J* = 3.2 Hz), 126.09, 124.24, 122.27, 122.19, 118.35, 115.49 (C_2_, *J* = 22.0 Hz), 112.04, 109.81, 103.51, 100.02, 66.72, 50.37, 48.19, 26.33, 23.73. Anal. calcd. for C_23_H_24_BrFN_2_O × H_2_O: C, 59.88; H, 5.68; N, 6.07. Found: C, 59.80; H, 5.63; N, 6.09.

(*R*)-4-[4-(1-phenylethylamino)butoxy]-9*H*-carbazole hydrobromide **9**

From carbazole **11** (0.16 mmol), compound (*R*)-**9** (0.053 g, 77%) was obtained as an amorphous yellowish solid; m.p. 92–96 °C. IR (KBr, cm^−1^) ν_max_: 3405, 3284, 2936, 2781, 1625, 1604, 1585, 1506, 1454, 1440, 1385, 1346, 1332, 1303, 1261, 1214, 1096, 786, 754, 724, 700. ^1^H NMR (600 MHz, CD_3_OD): δ = 8.22 (d, *J* = 7.7 Hz, 1H), 7.46–7.44 (m, 1H), 7.39–7.34 (m, 6H), 7.30 (t, *J* = 8.0 Hz, 1H), 7.18–7.14 (m, 1H), 7.09–7.07 (m, 1H), 6.66 (d, *J* = 7.9 Hz, 1H), 4.32–4.27 (m, 2H), 4.13 (q, *J* = 7.1 Hz, 1H), 3.12–3.06 (m, 1H), 2.93–2.87 (m, 1H), 2.10–1.95 (m, 4H), 1.62 (d, *J* = 6.8 Hz, 3H). ^13^C NMR (151 MHz, CD_3_OD): δ = 154.97, 141.57, 139.36, 136.54, 129.13, 129.03, 127.10, 126.07, 124.25, 122.26, 122.19, 118.36, 112.03, 109.82, 103.51, 100.00, 66.65, 58.13, 45.55, 26.18, 23.60, 18.35. Anal. calcd. for C_24_H_27_BrN_2_O × H_2_O: C, 63.02; H, 6.39; N, 6.12. Found: C, 62.82; H, 6.20; N, 5.98.

(*S*)-4-[4-(1-phenylethylamino)butoxy]-9*H*-carbazole hydrobromide **10** (enantiomer of **9**)

From carbazole **11** (0.16 mmol), compound (*S*)-**10** (0.066 g, 96%) was obtained as a yellowish oil. Anal. calcd. for C_24_H_27_BrN_2_O × H_2_O: C, 63.02; H, 6.39; N, 6.12. Found: C, 63.31; H, 6.34; N, 6.00.

### 4.2. Determination of Antimicrobial Activity

The antibacterial and antifungal properties of ten 4-[4-(benzylamino)butoxy]-9*H*-carbazole derivatives were investigated using the broth microdilution method, according to the Clinical and Laboratory Standards Institute (CLSI) norms for the antimicrobial susceptibility testing of bacteria that grow aerobically (M07, 11th Edition), and for the antifungal susceptibility testing of filamentous fungi (M38, 3rd Edition) and yeast (M27, 4th Edition). The antimicrobial activity was tested against Gram-positive (*Staphylococcus epidermidis* ATCC 12228 and *Staphylococcus aureus* ATCC 29213) and Gram-negative (*Escherichia coli* ATCC 25922 and *Pseudomonas aeruginosa* ATCC 15442) bacteria, as well as *Aspergillus flavus* ATCC 9643 and *Candida albicans* ATCC 10231 fungi. The growth of the microorganisms untreated and treated with the tested compounds was determined in 96-well cell culture plates in MH or RPMI medium for the bacteria and fungi, respectively. The antibacterial potential of 4-[4-(benzylamino)butoxy]-9*H*-carbazole derivatives was tested in the concentration range of 0.0625–64 µg/mL. The tested compounds were dissolved in DMSO, and then the stock solutions were diluted in an MH or RPMI medium. Ciprofloxacin and amphotericin B were tested as the positive control compounds in the antibacterial and antifungal activity assays, respectively. Ciprofloxacin and amphotericin B were dissolved in water (with 0.1 N HCL) and DMSO. The antibiotic activities were examined in the concentration range of 0.0635–64 µg/mL. The inoculums of the tested bacterial strains were diluted in an MH medium, and the final density of suspensions achieved the value of 5 × 10^5^ CFU/mL. The inoculum of *C. albicans* yeast was diluted in RPMI medium, to achieve a final suspension density of 2.5 × 10^3^ CFU/mL. The *A. flavus* spores were washed off the slant with RPMI medium, and counted with a Thoma counting chamber. The final density of spores was 2.5 × 10^4^ spores/mL. The bacterial cultures supplemented with the tested compounds were incubated for 24 h at 37 °C in the dark. In the case of the fungal cultures, the incubation lasted for 48 h at 37 °C in the dark. Appropriate abiotic controls and microbial growth controls were prepared in a manner analogous to the samples with the addition of the tested compounds, and they were incubated under the same conditions. The optical density of all the samples was measured at λ = 630 nm, using a microplate reader (Multiskan FC Microplate Photometer, ThermoFisher Scientific, Pudong, Shanghai, China). The growth of the tested microorganisms was calculated as percentages of the control, based on the values of optical density from four experiments (n = 4). The minimum inhibitory concentrations (MICs) of the tested compounds were defined as the lowest concentrations of the compounds at which no growth in bacteria or fungi was observed. The minimum bactericidal concentrations/minimum fungicidal concentrations (MBCs/MFCs) were designed as the lowest concentrations that completely limited the cell viability, and they were determined after the microbial suspensions were placed on an agar medium. The values of MICs, MBCs, and MFCs are expressed in µg/mL.

### 4.3. Evaluation of the Hemolytic Activity of 4-[4-(benzylamino)butoxy]-9H-carbazole Derivatives

Red blood cells (RBCs) were purchased from the Regional Centre of Blood Donation and Blood Treatment in Lodz (Poland). The hemolytic properties of the tested derivatives were assessed based on spectrophotometric measurement of the released hemoglobin content.

RBCs were resuspended in PBS to obtain a hematocrit of 2.5%, after the cells were washed three times in PBS. The hemolytic activity of 4-[4-(benzylamino)butoxy]-9*H*-carbazole derivatives was examined at the concentrations of 1–64 µg/mL. The RBCs suspended in PBS, without the addition of the tested compounds and suspended in sterile pure deionized water, were prepared as negative and positive control samples, respectively. All the samples and controls were incubated in the dark at 37 °C for 24 h and, after that, the test tubes were centrifuged at 2800 rpm for 20 min, and the absorbances of the supernatants were measured spectrophotometrically at λ = 540 nm, using a MultiskanTM FC Microplate Photometer (Thermo Fisher Scientific, Pudong, Shanghai, China). The hemolytic potential of the tested compounds was calculated using the following formula:% Hemolysis = As/Ac × 100%
where As was the absorbance of samples supplemented with the tested compounds, and Ac was the absorbance of the positive control without the compounds.

The hemolytic activity was expressed as a percentage of the hemolysis in the positive control samples from four experiments (n = 4).

### 4.4. Determination of the Cytotoxic Activity of 4-[4-(benzylamino)butoxy]-9H-carbazole Derivatives

The cytotoxic properties of the tested compounds were examined using the human fibroblast BJ ATCC CRL-2522 cell line. The fibroblasts, with a final density of 1 × 10^5^ cells per well, were suspended in EMEM medium containing FBS (10%) and antibiotics (penicillin, 100 IU/mL and streptomycin, 100 µg/mL), and incubated in 96-well microplates for 24 h in a humidified atmosphere with 5% CO_2_ at 37 °C. Next, the EMEM medium was replaced with a fresh medium, with the addition of the tested compounds in the concentration range of 1–64 µg/mL. Cell cultures without the tested derivatives were also prepared, as control samples. All the samples were incubated again for 24 h, under the same conditions. Then, the growth medium was removed from above the fibroblasts, and MTT solution (500 µg/mL) was added to each well. The plates were incubated for 4 h under the same conditions. The MTT solution was removed, and the wells were refilled with 50 µL of DMSO, to dissolve the formazan crystals produced. The spectrophotometric measurement of the samples was carried out at λ = 550 nm, using a SpectraMax i3x Multi-Mode Microplate Reader (Molecular Devices Ltd., Wokingham, UK).

The cell viability was calculated as a percentage of the control samples untreated with the tested compounds with a standard deviation. The experiment with n = 4 was carried out in four independent repetitions.

### 4.5. Statistical Analysis

All results were presented as the means with a standard deviation (SD). One-way analysis of variance (ANOVA) at *p* < 0.05 was used to consider the statistical significance of the results. The analyses were prepared using Excel, and Microsoft Office 365 Business Premium (Microsoft Corporation, Redmond, WA, USA).

## 5. Conclusions

The purpose of our research was to investigate the cytotoxicity and antimicrobial activity of newly synthesized isomeric and fluorinated 4-[4-(benzylamino)butoxy]-9*H*-carbazole derivatives. These new compounds showed significant potential against a Gram-positive *S. aureus* strain. Complete bacterial growth inhibition was observed in the *S. aureus* cultures incubated with carbazole derivatives **2**–**5** and **7**–**10** at the concentration of 32 µg/mL. The addition of compounds **2**, **4**, and **8** at the concentration of 16 µg/mL caused a reduction of over 60% in the growth of the microorganism. It appears that the examined structures may be effective antimicrobial agents. Functionalized 4-[4-(benzylamino)butoxy]-9*H*-carbazole derivatives proved to be more effective than the parent compound. Therefore, the tested derivatives may be promising structures for further functionalization, in order to increase their antimicrobial activity, and decrease their cytotoxicity.

## Data Availability

The data presented in this study are available on request from the corresponding author.

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
