# Peer review of "Antimicrobial Activity and Toxicity of Newly Synthesized 4-[4-(benzylamino)butoxy]-9H-carbazole Derivatives"

_ijms, 2023, doi:10.3390/ijms241813722_

Round 1

Reviewer 1 Report

Dear authors,

I read the manuscript with interest.

The work is very interesting, but there are some points that should be fixed before publication in International Journal of Molecular Sciences

1. The abstrat should be improved according to the results

2. The introductory part is very lengthy and needs much editing with suitable references.

3. The synthetic route is already described in the literature, what is new in this synthesis?

4. It will be better if you can add some theoretical studies as molecular docking and dynamics.

5. The quality of figures ans schemes should be improved.

The English language should be improved.

Author Response

Thank you for giving us the opportunity to submit a revised draft of our manuscript entitled “Antimicrobial activity and toxicity of newly synthesized 4-[4-(benzylamino)butoxy]-9H-carbazole derivatives” to the International Journal of Molecular Sciences (ijms- 2542923). We appreciate the time and effort the Reviewers has dedicated to providing your valuable feedback on our manuscript. We would like to thank the Reviewers for the interest in our work and valuable comments that helped to improve the manuscript. The changes have been marked in the change tracking mode. Here are the point-by-point responses to the Reviewer comments and concerns.

1.The abstrat should be improved according to the results

Answer: The abstract has been extended with a short description of the obtained results.

2. The introductory part is very lengthy and needs much editing with suitable references.

Answer: The introduction part has been shortened.

3. The synthetic route is already described in the literature, what is new in this synthesis?

Answer:  indeed, the previously described procedure was applied for the synthesis of the designed compounds, however neither amines 1–5 nor hydrobromides 6–10 have been obtained so far. Therefore, in this manuscript full characteristic of all newly synthesized compounds is presented.

4. It will be better if you can add some theoretical studies as molecular docking and dynamics.

Answer: The description of the molecular docking of the carazole-oxadiazoles has been included in the Discussion section.

5. The quality of figures ans schemes should be improved.

Answer: Figures and diagrams have been enlarged and their quality has been improved.

Reviewer 2 Report

Scheme 1 (77-81) requires explanation. The reaction conditions "b" involves two inorganic anions and allows formation of two different amine salts: hydrobromide and hydroiodide, therefore procedure 4.1.2. should be supplemented with analytical determination of both anions in the isolate.

Compounds 1-5 and 6-10 are pharmacodynamically equivalent in view of the large buffering capacity of an organism treated for a microbial infection. The reason for considering the same API as two subsets should be explained with a reference to the principles of the pharmaceutical salts application.

Author Response

Thank you for giving us the opportunity to submit a revised draft of our manuscript entitled “Antimicrobial activity and toxicity of newly synthesized 4-[4-(benzylamino)butoxy]-9H-carbazole derivatives” to the International Journal of Molecular Sciences (ijms- 2542923). We appreciate the time and effort the Reviewers has dedicated to providing your valuable feedback on our manuscript. We would like to thank the Reviewers for the interest in our work and valuable comments that helped to improve the manuscript. The changes have been marked in the change tracking mode. Here are the point-by-point responses to the Reviewer comments and concerns.

Scheme 1 (77-81) requires explanation, The reaction conditions “b” involves two inorganic anions and allowes formation of two amine salts: hydrobromide and hydrochloride, therefore procedure 4.1.2. should be supplemented with analytical determination of both anions in the isolate.

Answer: We are grateful for the reviewer’s comment.

Indeed, the obtained compounds were analyzed using a standard analytical test with chlorine water for the presence of bromine and/or iodide anions which proved the presence of respective hydrobromides 6–10 only.

The brownish color of the tested solution was observed, while the violet coloring of the chloroform layer was not observed.

Moreover, all newly synthesized compounds are fully characterized. Elemental analyses for all compounds are given. All spectral data proved not only the structure but also the purity of the newly synthesized compounds which was finally confirmed by elemental analyses, we feel that the collected data sufficiently support structural assignments for all obtained compounds. So, we kindly ask the reviewer to accept the compounds’ characterization in their current form.

Compounds 1-5 and 6-10 are pharmacodynamically equivalent in view of the large buffering capacity of an organism treated for a microbial infection. The reason for considering the same API as two subsets should be explained with a reference to the principles of the pharmaceutical salts application.

Answer: An application of pharmaceutical salt has been developed in the Discussion section.

Reviewer 3 Report

The present manuscript by Katarzyna NiedziaÅ‚kowska et al describes the preparation of 10 benzylamino-butoxy-carbazole derivatives (5 free bases and 5 respective hydrobromides) and their  in vitro study against pathogenic microorganisms. A number of the tested compounds exhibited moderate activity against two strains of Gram positive bacteria. The synthesized compounds did not show significant effects toward red blood cells. However, they were more toxic to human fibroblasts than to erythrocytes. Overall, the results could be of interest to researchers in the field. In revising their manuscript the authors should include a positive control compound to all the biological tests, and results should be appropriately presented. Furthermore, the type of the instruments used to obtain the spectra (IR & NMR) and especially the elemental analyses should be reported.

Minor editing of English language required

Author Response

Thank you for giving us the opportunity to submit a revised draft of our manuscript entitled “Antimicrobial activity and toxicity of newly synthesized 4-[4-(benzylamino)butoxy]-9H-carbazole derivatives” to the International Journal of Molecular Sciences (ijms- 2542923). We appreciate the time and effort the Reviewers has dedicated to providing your valuable feedback on our manuscript. We would like to thank the Reviewers for the interest in our work and valuable comments that helped to improve the manuscript. The changes have been marked in the change tracking mode. Here are the point-by-point responses to the Reviewer comments and concerns.

The present manuscript by Katarzyna NiedziaÅ‚kowska et al describes the preparation of 10 benzylamino-butoxy-carbazole derivatives (5 free bases and 5 respective hydrobromides) and their in vitro study against pathogenic microorganisms. A number of the tested compounds exhibited moderate activity against two strains of Gram positive bacteria. The synthesized compounds did not show significant effects toward red blood cells. However, they were more toxic to human fibroblasts than to erythrocytes. Overall, the results could be of interest to researchers in the field. In revising their manuscript the authors should include a positive control compound to all the biological tests, and results should be appropriately presented. Furthermore, the type of the instruments used to obtain the spectra (IR & NMR) and especially the elemental analyses should be reported.

Answer: A positive control compounds were, such as commercial antibiotics, were tested and the data obtained were included in the Results section. Information about the type of the instruments used to obtain the spectra (IR & NMR) as well as for the elemental analyses were added as required.

Minor editing of English language required.

Answer: Another linguistic correction has been made. The certificate is also attached

Round 2

Reviewer 3 Report

The authors answered satisfactorily to all my questions/suggestions. Thus, I recommend publication of their manuscript without any further revision.

Author Response

We would like to thank the Reviewer for all valuable comments and acceptance of our answers.